# PTSD Risk Factors and Acute Pain Intensity Predict Length of Hospital Stay in Youth after Unintentional Injury

**DOI:** 10.3390/children9081222

**Published:** 2022-08-12

**Authors:** Anna Monica Agoston, Amina Bhatia, John C. Bleacher, Alexis Smith, Karen Hill, Susanne Edwards, Alicia Cochran, Maia Routly

**Affiliations:** 1Center for Pain Relief, Children’s Healthcare of Atlanta, Atlanta, GA 30322, USA; 2Division of Pediatric Anesthesiology, Emory University, Atlanta, GA 30322, USA; 3Division of Surgery, Emory University, Atlanta, GA 30322, USA; 4Division of Trauma Services, Children’s Healthcare of Atlanta, Atlanta, GA 30322, USA

**Keywords:** pain, PTSD, injury, hospitalization

## Abstract

**Background**: Many hospitals have adopted screening tools to assess risk for posttraumatic stress disorder (PTSD) after pediatric unintentional injury in accordance with American College of Surgeons recommendations. The Screening Tool for Early Predictors of PTSD (STEPP) is a measure initially developed to identify youth and parents at high risk for meeting diagnostic criteria for PTSD after injury. Acute pain during hospitalization has also been examined as a potential predictor of maladaptive outcomes after injury, including PTSD. We investigated in a retrospective cohort study whether the STEPP, as well as acute pain intensity during hospitalization, would predict maladaptive outcomes during the peri-trauma in addition to the post-trauma period, specifically length of hospitalization. **Methods**: A total of 1123 youths aged 8–17 (61% male) and their parents were included. Patients and parents were administered the STEPP for clinical reasons while hospitalized. Acute pain intensity and length of stay were collected through retrospective chart review. **Results**: Adjusting for demographics and injury severity, child but not parent STEPP total predicted length of stay. Acute pain intensity, child threat to life appraisal, and child pulse rate predicted length of stay. **Conclusions**: Acute pain intensity and child PTSD risk factors, most notably child threat to life appraisal, predicted hospitalization length above and beyond multiple factors, including injury severity. Pain intensity and child appraisals may not only serve as early warning signs for maladaptive outcomes after injury but also indicate a more difficult trajectory during hospitalization. Additional assessment and treatment of these factors may be critical while youth are hospitalized. Utilizing psychology services to support youth and integrating trauma-informed care practices during hospitalization may support improved outcomes for youth experiencing unintentional injury.

## 1. Background

Around 20 million children and adolescents in the United States are affected by unintentional injury, with one in five deaths due to this cause [1] and 29 hospitalizations occurring per each death [1,2]. There has been significant concern around the development of posttraumatic stress disorder (PTSD) after injury and prevalence rates are between 20–25% [3,4,5]. PTSD is associated with negative outcomes post-trauma and is linked to poorer functional health outcomes and greater utilization of health services [6,7,8]. Predictors of PTSD in youth experiencing unintentional injury have been extensively studied to enhance screening and prevention efforts. Evidence-based screening tools assessing risk for PTSD are increasingly being administered within hospitals in line with American College of Surgeons recommendations for assessing and addressing this risk [9]. Acute pain intensity during hospitalization has also been investigated as a predictor of PTSD [10,11] and other maladaptive outcomes such as chronic post-surgical pain [12], although limited research has examined these relations.

Although PTSD screening tools are designed for predicting outcomes well into the post-trauma period (i.e., at least a month after initial injury considering PTSD diagnostic criteria), limited research has investigated the utility of applying information from these tools to capture maladaptive peri-trauma processes, which may contribute to risk for negative outcomes post-discharge. Risk factors for PTSD, such as physiological arousal or negative child appraisals, may affect youths’ ability to comply with medical care [13] and correlate with traumatic reactions to medical intervention, potentially reducing compliance with procedures and increasing length of hospitalization. Acute pain intensity during hospitalization may also reflect poorer pain control, which may result in additional medical involvement and delayed discharge. Hospital length of stay (LOS) after accounting for injury severity may be a useful proxy for capturing these peri-trauma processes. Utilizing real-time measures from the patient’s hospitalization (e.g., PTSD screening scores and acute pain intensity scores) may enable clinicians to provide increased support for patients during the peri-trauma period. In turn, this support may mitigate any additional traumatization experienced by the patient due to hospitalization and medical intervention, thereby potentially reducing the risk of PTSD post-trauma.

The current study aimed to examine the predictive value of risk factors for PTSD, including acute pain intensity, during the peri-trauma period. We hypothesized that higher risk for PTSD, including endorsement of maladaptive cognitive appraisals and higher acute pain intensity, would predict increased LOS. Parental appraisals may also be influential in impacting a child’s course of hospitalization and LOS through shaping youths’ appraisals and coping [14]. Parental presence has been found to decrease youths’ intensity of pain during painful medical procedures [15], affecting their recovery. In light of research demonstrating the impact of parental functioning and parent–child interactions during the peri-trauma period, we examined parental risk factors for PTSD and hypothesized that higher parental risk would predict increased LOS.

## 2. Materials and Methods

This study utilized a retrospective electronic chart review to collect data. All participants were hospitalized for unintentional injury. Data were collected from two large pediatric hospitals in the southeastern region of the United States designated Level I and Level II trauma centers (the highest levels of care). All questionnaires and assessments were administered by nurses. All study procedures were approved by the hospital Institutional Review Board.

Participants included 1123 youths (median age: 13.02, IQR: 3.95). Data were collected from March 2018 to December 2021. Inclusion criteria included any participant who had sustained a physical injury requiring hospitalization and were between the ages of 8–17 at admission. Exclusion criteria included brain injury that impeded the ability of the participant to respond to questions adequately, intubation throughout the hospitalization, and significant developmental delay. Parents and youth were not required to have adequate English language proficiency; if their English was not proficient, an interpreter was used.

### 2.1. Assessment of Risk Factors for PTSD

The Screening Tool for Early Predictors of PTSD (STEPP) is a brief screening tool for youth experiencing unintentional injury to identify risk for PTSD in youth and parents [16]. The STEPP is validated for youth between the ages of 8–17 and was originally derived from a survey of youth and parents querying evidence-based and theoretical domains associated with the development of PTSD [16]. Four questions are asked of the parents (i.e., Did you see the incident in which your child got hurt? Were you with your child in an ambulance or helicopter on the way to the hospital? When your child was hurt, did you feel really helpless, like you wanted to make it stop happening, but you couldn’t? Does your child have any behavior problems or problems paying attention?), four of the child (i.e., Was anyone else hurt or killed? Was there a time when you didn’t know where your parents were? When you got hurt, or right afterwards, did you feel really afraid? When you got hurt, or right afterwards, did you think you might die?), and then three items are recorded from the medical record (i.e., Suspected extremity fracture? Was pulse rate at emergency department triage > 104/min if the child is under 12 years or >97/min if the child is 12 years or older? Is the child 12 years or older? Is this a girl?). These items are dichotomously coded (1 for Yes, 0 for No). Child screens are positive when 4 out of 8 total items are endorsed, and parent screens are positive when 3 out 6 total items are endorsed. A positive screen indicates a likelihood of developing future PTSD. Studies examining rates of positive screens have been limited and range widely [16,17,18] despite similar demographics and methods across studies. The STEPP has been reported to have high sensitivity (0.88) and moderate specificity (0.48) when detecting 6-month PTSD in the original study [16] but replications of this initial study have been mixed [17,18,19]. In light of concerns regarding the appropriate cutoff for the STEPP [17,19], we examined the STEPP as a dimensional (i.e., continuous) variable in these analyses. The STEPP was administered as close to the start of admission as possible by a limited pool of trauma nurses based on nurse availability and accessibility to patient (e.g., patient was out of surgery, not completing other medical tests/procedures, had been extubated if intubated). The median time elapsed between admission and administration of STEPP was 0.80 days.

### 2.2. Acute Pain Intensity

During the patient’s hospitalization, nurses asked patients “what is your current pain level” using a numeric rating system, with 0 reflecting no pain and 10 reflecting the worst pain imaginable. The highest score recorded across the duration of hospitalization was used as the acute pain intensity score. The numeric rating system for pain is considered valid and reliable for ages 8 to 18 [20]. The median time elapsed between admission and collection of the acute pain intensity score was 1.29 days.

### 2.3. Length of Stay

LOS was collected from medical records based on how many days were recorded between admission and discharge. Patients considered under observation were included in the analyses if they were being observed on a medical floor.

### 2.4. Injury Severity Score

The Injury Severity Score standardizes the severity of traumatic injury by providing a total of scores of the worst injury within six categories of body systems [21]. This total score demonstrates better prognostic utility for predicting outcomes in youth than other commonly used scores such as the Glasgow Coma Score and the Pediatric Trauma Score [22]. This total score was calculated by trauma staff and entered into a trauma database from which these scores were obtained for the current study.

### 2.5. Statistical Analyses

Data were analyzed using SPSS Version 28.0, IBM, Armonk, NY, USA. Descriptive statistics were used to summarize socio-demographic factors as well as characteristics of the variables in the study. T-tests were used to examine any sex, age, or race differences within the variables in our sample. Bivariate correlations were used to examine relations between variables of interest, particularly correlations between acute pain intensity, LOS, and individual items on the STEPP.

We conducted a series of hierarchical multivariable regression analyses to examine child/parent STEPP totals and items predicting LOS, as well as acute pain intensity predicting LOS. We examined totals separately and items separately first. We then combined all items and acute pain intensity into one regression and mean-centered predictors/covariates before entry to reduce collinearity as suggested by Aiken and West [23].

Considering that multiple factors may influence these relationships, including demographics [24] and injury severity [25], we adjusted for age, sex, race, ethnicity, and injury severity score. Race was specifically examined in light of racial and ethnic disparities in trauma exposure [26] and pain management [27]. Covariates were entered at the first step and predictors were entered at the second step.

## 3. Results

### 3.1. Descriptive Analyses

Table 1 presents socio-demographics of the sample and variable descriptives.

Approximately two-thirds of participants were male (61.0%). Slightly more than half of participants identified as white (51.6%), with the rest of participants identifying mostly as black (40.1%). Almost a third of injuries occurred from motor vehicle accidents (28.9%), with the remaining being recreational activities, organized sports, dog bites, pedestrian injured by automobile, or gunshot wounds. The injury severity score ranged from 1–75 with a mean of 9.62 (*SD* = 7.16). A total of 465 patients were admitted under observation to the medical floor and 658 were classified as inpatients.

### 3.2. Sex, Age, and Race Differences among Variables

A series of *t*-tests was conducted to examine sex, age, and race differences (white vs. black).

Compared to males, females showed significantly higher acute pain intensity (*M* = 7.91, *SD* = 2.27 vs. *M* = 7.57, *SD* = 2.43, *t* = 2.36, *p* = 0.009, *d* = 0.14) and child STEPP total after removing the sex item (*M* = 2.30, *SD* = 1.39 vs. *M* = 1.91, *SD* = 1.40, *t* = 4.55, *p* < 0.001, *d* = 0.34). No significant sex differences were found for LOS, injury severity, or parent STEPP total.

Compared to younger participants, older participants showed significantly higher acute pain intensity (*M* = 7.84, *SD* = 2.28 vs. *M* = 7.54, *SD* = 2.47, *t* = 2.10, *p* = 0.04, *d* = 0.13). Compared to older participants, younger participants showed significantly higher child STEPP totals (*M* = 2.76, *SD* = 1.49 vs. *M* = 2.23, *SD* = 1.53, *t* = 5.87, *p* < 0.001, *d* = 0.35). It was unclear the pattern for parent STEPP total because older participants had a higher parent STEPP total (*M* = 2.31, *SD* = 0.94 vs. *M* = 1.85, *SD* = 1.04, *t* = 7.29, *p* < 0.001, d = 0.46) but then younger participants did when taking out the age item (*M* = 1.65, *SD* = 1.01 vs. *M* = 1.35, *SD* = 0.91, *t* = 4.76, *p* < 0.001, *d* = 0.31). No significant age differences were found for LOS or injury severity score.

Compared to white participants, black participants demonstrated a significantly higher child STEPP total (*M* = 2.63, *SD* = 1.49 vs. *M* = 2.41, *SD* = 1.57, *t* = 2.28, *p* = 0.02, *d* = 0.14). Compared to black participants, white participants showed a significantly higher injury severity score (*M* = 10.18, *SD* = 7.50 vs. *M* = 8.82, *SD* = 6.68, *t* = 2.73, *p* = 0.007, *d* = 0.19). No significant race differences were found for acute pain intensity, LOS, or parent STEPP total.

### 3.3. Correlations among Variables

Table 2 presents intercorrelations among the variables. Significant correlations were found between acute pain intensity, LOS, and multiple STEPP parent and child items.

### 3.4. Hierarchical Multiple Regression Analyses

We first investigated demographic variables and injury severity as predictors of LOS.

Injury severity score significantly predicted LOS (*ß* = 0.35, *t* = 11.38, *p* < 0.001) such that higher injury severity score was associated with longer LOS. All other demographic variables were nonsignificant. After adjusting for the variables outlined within the statistical analyses section, regressions revealed higher child STEPP total significantly predicted longer LOS (*ß* = 0.15, *t* = 4.21, *p* < 0.001); however, there were no associations between parent STEPP total and LOS (*ß* = 0.01, *t* = 0.25, *ns*). When all of the child STEPP items were investigated together, threat to life appraisal and child pulse rate significantly predicted LOS (*ß* = 0.12, *t* = 3.56, *p* < 0.001 and *ß* = 0.12, *t* = 3.58, *p* < 0.001, respectively). None of the parent STEPP items were predictive of LOS (*ß* = −0.06–0.06, *t* = −1.59–1.55, *ns*) with one exception: parental helplessness predicted LOS when we removed *both* acute pain intensity and child threat to life appraisal in the analyses (*ß* = 0.09, *t* = 2.60, *p* = 0.01). All of the predictions for the items were significant in a positive direction so endorsement of the item was associated with longer LOS. Acute pain intensity significantly predicted LOS (*ß* = 0.22, *t* = 6.73, *p* < 0.001) such that higher acute pain intensity predicted longer LOS. When parent and child STEPP items were combined into one regression with acute pain intensity, the significance of results was the same as within the separate parent–child analyses. The results of the combined regression analysis are presented in Table 3.

## 4. Discussion

In the current study, acute pain intensity, as well as child risk factors for PTSD, predicted LOS above and beyond multiple factors, including demographics and injury severity. Specifically, child threat to life appraisal and pulse rate were both highly predictive of LOS. Our results are in line with a biopsychosocial model proposed by Marsac and colleagues [28] in which early child cognitive appraisals of the event leading to injury affect the peri-trauma period and development of PTSD symptoms. These results are also consistent with previous studies demonstrating correlations between acute pain intensity and LOS in adults [29,30], although previous research has been mixed in this area [31].

Threat to life appraisal may result in increased physiological arousal [13], reflected in pulse rate shortly after injury as well as increased pain [32]. Acute pain may trigger hypothalamic-pituitary-adrenal and noradrenergic activation that enhances fear conditioning, increasing the risk of overconsolidation of trauma-related memories [33] and putting youth at risk of prolonged hospitalization [34]. Negative and trauma-related appraisals may continue during hospitalization and correlate with traumatic reactions to medical procedures or other medical experiences (e.g., insertion of IV, dressing changes), which may reduce compliance with procedures and increase utilization of resources (e.g., multiple nurses may be required to calm patient), thereby lengthening time to discharge. Sleep may be disrupted by increased physiological arousal, leading to poorer compliance during the day with required therapies (e.g., physical therapy) and delayed discharge. Finally, parental helplessness predicted LOS but this association was no longer significant when acute pain intensity and child threat to life appraisal were included in the analyses. Although parental appraisals may be influential [14], acute pain intensity and child appraisals are more predictive in determining youths’ course of hospitalization.

Implementing interventions to address child appraisals and acute pain while youth are still hospitalized may be effective in mitigating negative outcomes during hospitalization and reducing LOS. When risk factors on the STEPP are present, assessment of early appraisals using tools such as those developed by Marsac and colleagues [14] may be warranted. Consulting psychologists and other mental health professionals to address maladaptive appraisals using cognitive behavioral techniques may help youth shift appraisals while hospitalized. Web-based programs such as AfterTheInjury.org [35] may be useful for helping parents facilitate recovery for youth. Pharmacological and nonpharmacological strategies such as deep breathing may address pain during hospitalization [36]. Administration of morphine during hospitalization led to reduced risk for PTSD on 6-month follow-up [32], suggesting adequate pharmacological treatment of pain may also improve peri-trauma outcomes. Finally, integrating trauma-informed care practices such as those outlined by Agoston and colleagues [37] may be critical. Namely, supporting sleep hygiene, engaging families in decision-making, scheduling daily living activities, and creating a healing environment around patients may significantly reduce LOS and utilization of medical resources, decreasing the cost of healthcare.

Our study indicates that child subjective (i.e., cognitive appraisals, pain) and objective (i.e., pulse rate) factors may not only affect risk for PTSD as indicated by previous studies but also affect the course of a child’s hospitalization. Longer hospitalizations may incur additional utilization of resources and magnify the financial burden associated with pediatric injury. Although our study did not explicitly examine the effects of child cognitions on medical costs after injury, LOS is likely highly correlated with higher financial burden and future studies may explicitly examine these links. Strengths of the study include a relatively large sample size. We also emphasize utilizing existing screening tools already implemented in many hospitals, minimizing the need for deployment of additional resources.

With regard to limitations, unfortunately no mechanism of action was investigated with regard to how pain or risk factors for PTSD affect youths’ LOS. Future studies may investigate these mechanisms. Our results also demonstrate there are multiple determinants of LOS that are not examined here, which may be investigated in future research. Notably, we did not measure acute stress symptoms in this study. While screening tools such as the STEPP utilize pre- and peri-trauma subjective and objective factors [16], others involve the assessment of acute stress symptoms immediately following a trauma [38]. Considering around 88% of both youth and parents reported at least one clinically significant acute stress symptom [39], these symptoms may fail to reliably distinguish individuals benefiting from additional intervention. Research also indicates that risk factors for PTSD other than acute stress symptoms (e.g., cognitive appraisals) may predict PTSD better than acute stress symptoms [40,41] and may even mediate the relationship between initial symptoms and PTSD well after injury [42]. Nevertheless, collecting information on acute stress symptoms may be helpful to elucidate the dynamics between risk for PTSD and PTSD symptoms themselves immediately after injury, as well as how these symptoms may contribute to delayed discharge.

In conclusion, this study illustrates how a common screening tool assessing risk for PTSD, as well as acute pain intensity, could be used to predict an important outcome (i.e., LOS) during the peri-trauma period. Addressing cognitive appraisals and pain while youth are hospitalized may be extremely useful in not only reducing risk for PTSD after injury but also decreasing LOS, thereby potentially minimizing the utilization of medical resources during hospitalization. Using psychology services within the hospital and integrating trauma-informed care practices may be critical in supporting improved outcomes for patients experiencing unintentional injury.

## Figures and Tables

**Table 1 children-09-01222-t001:** Participant Demographics and Descriptives.

Demographics	*N* (%)	*Min*	*Max*	*M*	*SD*
1. Age	1123	8.00	17.99	12.78	2.50
2. Sex	1123				
Female	438 (39.0)				
Male	685 (61.0)				
3. Race	1123				
White	579 (51.6)				
Black	450 (40.1)				
Asian	17 (1.5)				
American Indian/Native Hawaiian	3 (0.3)				
Biracial (White/Black)	16 (1.4)				
Declined	58 (5.2)				
4. Ethnicity	1120				
Hispanic	156 (13.9)				
Non-Hispanic	964 (86.1)				
5. Injury Severity Score	919	1.00	75.00	9.62	7.16
6. Child STEPP Total	1099	0	8	2.48	1.54
7. Parent STEPP Total	973	0	5	2.09	1.01
8. Acute Pain Intensity	1123	0	10	7.70	2.38
9. Length of Stay	1123	0.00	66.00	3.81	5.70

**Table 2 children-09-01222-t002:** Intercorrelations Among the Variables.

Variables	1	2	3	4	5	6	7	8	9	10	11	12	13	14
1. Acute Pain Intensity	--	0.22 ***	−0.02	−0.06 ^	0.16 ***	0.05 ^	0.10 **	0.10 **	0.07 *	0.15 ***	0.09 **	0.09 **	0.06 *	0.24 ***
2. Length of Stay		--	0.02	−0.07 *	0.12 ***	0.04	0.03	0.09 **	0.04	0.15 ***	0.13 ***	0.01	0.04	0.10 ***
3. See child hurt (P)			--	0.20 ***	−0.02	−0.13 ***	0.09 **	−0.07 *	0.01	0.01	0.03	0.05	−0.05	−0.04
4. Present in ambulance/helicopter (P)				--	−0.05	−0.06 *	−0.16 ***	0.00	0.07 *	0.04	−0.05	0.05	−0.01	−0.06 ^
5. Felt helpless (P)					--	−0.15 ***	0.10 **	0.12 ***	0.22 ***	0.23 ***	0.10 **	0.05	0.00	−0.06 ^
6. >12 years (P)						--	−0.11 ***	−0.02	−0.20 ***	−0.06 ^	−0.04	−0.04	0.00	0.00
7. Anyone else hurt/killed (C)							--	0.11 ***	0.06 ^	0.06 *	0.15 ***	0.19 ***	−0.05	0.08 *
8. Did not know where parents were (C)								--	0.16 ***	0.16 ***	0.04	0.01	0.13 ***	0.00
9. Felt afraid (C)									--	0.37 ***	0.03	0.12 ***	0.03	−0.04
10. Thought may die (C)										--	0.07 *	0.05 ^	0.16 ***	0.05 ^
11. Pulse Rate (C)											--	0.13 ***	0.05 ^	0.10 **
12. Girl (C)												--	−0.07 *	−0.03
13. Prior behavioral/attention issues (C&P)													--	−0.01
14. Suspected Extremity Fracture(C&P)														--

*Note.* C is child item. P is parent item. C&P are both child and parent items. ^ *p* < 0.10, * *p* < 0.05, ** *p* < 0.01, *** *p* < 0.001.

**Table 3 children-09-01222-t003:** Hierarchical Multiple Regressions Predicting Length of Stay.

Criterion: Length of Stay			
**Step 1**	Beta (Standardized)	*t*	Δ*R*^2^
Age	0.05	1.31	0.13 ***
Sex	−0.01	−0.25	
Race	0.02	0.46	
Ethnicity	0.05	1.28	
Injury Severity Score	0.36	10.38 ***	
**Step 2**			0.09 ***
** *STEPP Parent Items* **			
See child hurt	0.01	0.41	
Present in ambulance/helicopter	−0.05	−1.47	
Felt helpless	0.02	0.49	
>12 years	−0.06	−1.09	
** *STEPP Child Items* **			
Anyone else hurt/killed	0.00	−0.01	
Did not know where parents were	0.04	0.99	
Felt afraid	−0.01	−0.14	
Thought may die	0.12	3.12 **	
Pulse Rate	0.13	3.70 ***	
Girl	0.07	0.54	
** *STEPP Child and Parent Items* **			
Prior behavioral/attention issues	−0.01	−0.35	
Suspected Extremity Fracture	−0.01	−0.29	
** *Acute Pain Intensity* **	0.19	5.38 ***	

*Note.* ** *p* < 0.01. *** *p* < 0.001.

## Data Availability

Deidentified data available from study authors upon request for replication purposes.

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
