# Peer review of "PTSD Risk Factors and Acute Pain Intensity Predict Length of Hospital Stay in Youth after Unintentional Injury"

_children, 2022, doi:10.3390/children9081222_

Round 1

Reviewer 1 Report

Authors made a study named: “ PTSD Risk Factors and Acute Pain Intensity Predict Length of 2 Hospital Stay in Children After Unintentional Injury”.

My remarks:

1. Abstract: PTSD post-trauma is a term little reader unfriendly! Should be: PTDS after injury or something in that direction.

2. Abstract: authors state that this is a cohort study? Which type of cohort study- retrospective or prospective? What was done for this to be a cohort study?

3. Introduction: Final third of introduction is more a discussion section so should be places into discussion.

4. Methods: should write more for the description and details in data collecting.

5. Methods: what does the unintentional injury comprise all? bone trauma, soft tissue trauma, psychological trauma? Included patients were operated or just admitted, or both summarized and not divided into groups?

6.. Methods: what does the Level I and II mean? how many levels are there? Please explain.

7. Methods: the mean age should be shown as medians with IQR.

8. Methods: are poorly written, very reader unfriendly and not systematic. Should be better set, by normal order, should state clearly inclusion and exclusion criteria. And should be proof read.

9. Methods: Assessment of Risk Factors for PTSD- no need for repetitions from introduction.

10. Methods: Assessment of Risk Factors for PTSD- authors should describe, in details, the exact type of scoring they used.

The sentence: “Four questions are asked of the parents, four of the child, and then three items are rec-117 orded from the medical record.“ is not sufficient. What items? This can also be placed as an attachment.

11. Methods: Assessment of Risk Factors for PTSD- no need to discuss the specificity and sensitivity of the test here. Should be in discussion.

12. Methods: Acute Pain Intensity - for younger children, and this could be unified for the study; a visual analog scale (VAS) should have been used. Just as a suggestion, but being this was a retrospective study, or at least I think it was, and then the current model is only applicable...

13. Results: table 1 is shifted left, should be realigned.

14. Authors should at least add the limitations of the study, there are quite some.

It is hard for me to comprehend the comparison of these too heterogeneous and diversiform variables. Aside from the exact trauma cause, the authors should objectivize the trauma severity and compare somewhat unified patients. Additionally there is an issue where some of this children under general anesthesia, operation, what kind of operation, just a wound primary closure or/and osteosynthesis, exploration of abdominal cavity. This way we see a lot of hardly comparable variables, race, age, trauma cause, etc...

Although I concur with the conclusions, there methodological issues should be addressed. This way the results would be much accurate cause this type of investigation deals with a lot of subjectiveness.

Author Response

Response to Review

We really appreciate the thorough reviews and are thankful that both reviewers see this manuscript as a worthy contribution to Children. Below is a response to the Reviewer #1’s individual points:

Reviewer 1:

  1. Abstract: PTSD post-trauma is a term little reader unfriendly! Should be: PTDS after injury or something in that direction.

As suggested by the reviewer, we have replaced “PTSD post-trauma” with “PTSD after injury” in the abstract and reduced the use of the term “post-trauma” throughout the manuscript.

  1. Abstract: authors state that this is a cohort study? Which type of cohort study- retrospective or prospective? What was done for this to be a cohort study?

We are happy to clarify this point. This is a retrospective cohort study, which is now noted in the abstract.  The cohort is all participants that sustained an unintentional injury requiring hospitalization and were between the ages of 8-17 (ages that made them eligible to receive a STEPP screen). This is now noted in the eligibility criteria.

  1. Background: Final third of introduction is more a discussion section so should be places into discussion.

In line with the reviewer’s suggestion, we have made substantial edits to the introduction and moved a significant portion of this section to the discussion.

  1. Methods: should write more for the description and details in data collecting.

Thank you for this suggestion. We have now added additional details within the Method section about data collection (see edits throughout).

  1. Methods: what does the unintentional injury comprise all? Bone trauma, soft tissue trauma, psychological trauma? Included patients were operated or just admitted, or both summarized and not divided into groups?

The unintentional injury consists of all types of physical injury and includes both patients with and without operations. Due to operations having a wide range of severity (e.g., closure of wound vs. spinal fusion), as well as a wide range of severity for the initial trauma, we felt that separating into groups would be less than ideal. To address this concern, we included the Injury Severity Score to account for the difference in severity among initial injury as well as the likely subsequent level of operations needed (assuming initial injury may correlate with the level of repair needed for these injuries). To address the reviewer’s concern, we clarified this point in the description of the sample and included additional information about the Injury Severity Score in the section after Length of Stay. Notably, our goal for this study was to demonstrate utility of the STEPP screen and pain scores for predicting outcomes for all types of physical trauma.

  1. Methods: what does the Level I and II mean? how many levels are there? Please explain.

The Level I and II designation refers to the designated trauma center levels in the United States, which are determined by state law provisions.  Level I provides the highest level of surgical care to trauma patients, and Level II works in collaboration with a Level I institution. Level II and Level IV do not have the full availability of specialists and usually transfer to a Level I or Level II hospital. There are 4 levels in most states, including Georgia where the study was conducted. This point is clarified in the Methods section.

  1. Methods: the mean age should be shown as medians with IQR.

As suggested by the reviewer, we have included the median age and IQR in the Methods section Mean age and SD is still presented in Table 1.

  1. Methods: are poorly written, very reader unfriendly and not systematic. Should be better set, by normal order, should state clearly inclusion and exclusion criteria. And should be proof read.

We appreciate the feedback and have proof-read and restructured the methods. We have also clearly stated the inclusion and exclusion criteria.

  1. Methods: Assessment of Risk Factors for PTSD- no need for repetitions from introduction.

As Reviewer #2 had recommended, we removed all references to the STEPP screening tool in the Introduction and present information on the STEPP solely in the Methods section.

  1. Methods: Assessment of Risk Factors for PTSD- authors should describe, in details, the exact type of scoring they used.

As requested by both reviewers, we described the scoring of the STEPP within this section.

The sentence: “Four questions are asked of the parents, four of the child, and then three items are recorded from the medical record.“ is not sufficient. What items? This can also be placed as an attachment.

As recommended by both reviewers, we have included the individual items of the STEPP screen within the body of the text.

  1. Methods:Assessment of Risk Factors for PTSD- no need to discuss the specificity and sensitivity of the test here. Should be in discussion.

We were unclear regarding the reviewer’s rationale for moving specificity/sensitivity to the discussion. In light of previous precedent for including this section within the methods (e.g., Kassam-Adams et al., 2015) and because this was not a measure validation study, we chose to leave this section as is. We are open to moving it to the discussion if the reviewer may clarify the reason for doing so. 

  1. Methods: Acute Pain Intensity - for younger children, and this could be unified for the study; a visual analog scale (VAS) should have been used. Just as a suggestion, but being this was a retrospective study, or at least I think it was, and then the current model is only applicable...

We appreciate this suggestion. Our cohort is children 8-17 so the Numeric Rating Scale is valid for this cohort. If we do include a younger sample in future studies, we will definitely use the reviewer’s recommendation for an appropriate scale.

  1. Results: table 1 is shifted left, should be realigned.

This has been fixed in the manuscript.

  1. Authors should at least add the limitations of the study, there are quite some.

The limitations of this study had already been included in the second-to-last paragraph (before conclusions). However, in line with the reviewer’s point, we expanded our limitations section to provide additional thoroughness to this topic. In addition, we took a section from the introduction on screening of posttraumatic stress symptoms during hospitalization (now noted as acute stress symptoms) and discussed it as an additional limitation of the study. After reworking the Introduction, this section seemed more appropriate to note as a limitation.

  1. The authors should objectivize the trauma severity and compare somewhat unified patients.

We appreciate the author’s point that the participants’ injuries and hospital trajectory are heterogenous and diversiform. This is the reason why, as the reviewer suggested, we opted to adjust for objective trauma severity in our calculations. Separating into groups would limit our ability to generalize our findings to the larger trauma injury population. Considering that the STEPP is administered to the majority of trauma patients (with exclusion exceptions, see manuscript), we believed it was appropriate to draw conclusions about this larger sample as long as we accounted for the diversity of trauma and outcomes by including the Injury Severity Score. Point #5 (see above) further addresses these concerns.

Overall, we really appreciate the thoroughness of the reviewer’s suggestions and we hope that, with these changes, this reviewer will find our manuscript suitable for publication within Children’s.

Reviewer 2 Report

Title: PTSD Risk Factors and Acute Pain Intensity Predict Length of Hospital Stay in Children After Unintentional Injury

Thank you for the opportunity to review your paper “PTSD Risk Factors and Acute Pain Intensity Predict Length of Hospital Stay in Children After Unintentional Injury.” This very well written manuscript explores PTSD and pain factors that may impact length of hospital stay after pediatric unintentional injury. This paper makes a meaningful contribution to our understanding of these relationships and absolutely of interest to the readership. I thank the authors for engaging in this endeavor. I offer several mostly minor suggestions for improvement in clarity.

Overarching suggestions:

-        I recommend attending closely to the distinction between peri-trauma and post-trauma in the writing throughout the manuscript. This paper exclusively examines the peri-trauma outcome of hospital LOS, though there are many places where the authors refer to peri- AND post-trauma. There are also a few places where the authors refer to peri-trauma outcomes (plural vs. singular). When referring to their work in this study specifically, they should be clear on the scope/nature of what they examined/found.  

-        I recommend substituting “youth” for “children,” given that the sample is 8-17yo and includes adolescents. “Children” may mischaracterize the sample.

-        I recommend directing the reader to the tables throughout the manuscript text.

-        I recommend identifying a singular label with which to refer to each of the variables examined and being consistent with that label throughout the manuscript, including within the tables. For example, acute pain intensity is also referred to as maximum pain score and highest pain score. Please address for all variable names/labels.

Background:

-        Page 2, lines 52-69, it struck this reviewer as odd to be discussing the use of a specific measure within the introduction (given that this is not a measure validation paper), and that there is much speculation about the implications of using this tool/potential benefits before the authors have established any of their findings. I would encourage keeping the Background conceptual, discuss the measure in the Materials section, and save implications for the Discussion section.

-        Instead, the authors are encouraged to consider briefly touching upon some of the known risk factors for PTSD (i.e., those included in such screening tools) for important context. For instance, given the relevance of “child appraisals” in their findings later in the manuscript, I encourage the authors to define this (and other risk factors) before connecting them to LOS (lines 71-72). In other words, draw the link between child appraisal of the situation (e.g., thought may die) and PTSD, before connecting it to other outcomes.

Materials & Methods:

-        Page 3, lines 112-130, the authors are encouraged to:

-        Provide information on how the items are endorsed and scored (one would assume they are dichotomous as it is a screening tool) – are they coded 0/1?

-        Detail the specific items, given that these are examined individually. At the very least, the authors could refer the reader to table 2/3, but I would argue that more narrative information is needed about who answers which items, which come from the medical record, etc.

-        The authors are missing information about the measurement of “injury severity score.” How is this measured/rated? Who reports this/from where was this information gleaned? Is this a continuous variable?

Results:

-        Given that the variable “sex” is taken from the medical record, the assumption is that this refers to biological sex. If that is correct, it might make sense to report sex as “male/female” (vs. boy/girl). This recommendation is further relevant given the age range of the sample (8-17).

-        The authors use “child total” and “parent total” to refer to the respective total scores on the STEPP, but these labels alone lack clarity for the reader. Please reword as “child STEPP total” and “parent STEPP total” throughout this section/the manuscript. 

-        Page 4, line 168, please clarify what is meant by peds vs. auto.

-        Page 4, line 173, please describe the rationale for using a t-test to assess relationships with age, as is a continuous variable and you lose variability this way.

-        Page 5, line 189, please clarify that this finding was non-significant and remove the word marginally.

-        Page 5, please do not include t values for non-significant results.

-        In Table 2, the authors present correlations between continuous variables (LOS, pain scores) and dichotomous variables (STEPP items); were this point-biserial correlations to account for the categorical/dichotomous variables? Similarly, what was the rationale for examining intercorrelations between two dichotomous variables (items 3-14) vs. a chi-square?

-        Page 5, lines 202-203, this sentence should be reviewed and perhaps reworded for clarity. The double “as” makes it confusing, and it seems that there is just one DV (LOS) so variable should be singular.

-        Page 6, lines 212-213, please explain the rationale for removing pain score and threat to life appraisal from the analyses.

Discussion:

-        Broadly, the authors are encouraged to temper their discussion of the study’s implications to the scope of what was examined in their study. In other words, they should clearly distinguish between 1) direct implications of the findings and 2) future directions of study (e.g., while there is utility in examining the impact of these variables on post-trauma or peri-trauma outcomes other than LOS, these are future directions, rather than implications of this study).

-        Given that max pain, pulse rate, and appraisals contributed just 9% of the variance in LOS (beyond demographics and injury severity), the authors are encouraged to acknowledge that there are likely many other variables contributing to LOS that are not examined here. 

Author Response

Response to Review

We really appreciate the thorough reviews and are thankful that both reviewers see this manuscript as a worthy contribution to Children. Below is a response to the Reviewer #2’s individual points:

Reviewer 2:

We really appreciate the thorough reviews and are thankful that both reviewers see this manuscript as a worthy contribution to Children. Below is a response to the Reviewer #2’s individual points:

Overarching suggestions:

  1. Distinguish between peri-trauma and post-trauma in the writing throughout the manuscript.

We have made these changes throughout the manuscript. We clarified the text so that peri-trauma outcomes refer to LOS, and post-trauma refers to PTSD and other outcomes.  We also included additional clarification on the scope of our study being limited to peri-trauma processes and not post-trauma outcomes. We substantially reworked the entire discussion to limit the scope/nature of our findings so that the direct implications are clear and not confounded by future directions (see point #16, below for further clarification).

  1. Substitute “youth” for “children.”

As recommended by the reviewer, we have made this change throughout the manuscript.

  1. Direct the reader to the tables throughout the manuscript text.

We appreciate this suggestion. We believed we have already referred to tables throughout the manuscript text (see under Results – Descriptive Analyses, Correlations Among Variables, and last line of Hierarchical Regression Analyses). We are open to adding additional referrals if the reviewer feels like this is not adequate.

  1. Consistent labeling of variables, particularly acute pain intensity.

As recommended by the reviewer, we have consistently labeled the pain variable as “acute pain intensity” and removed any references to “maximum pain score” or “highest pain score.”

  1. Background. Keep the Background conceptual (e.g., briefly touch on known risk factors for PTSD), discuss the STEPP measure in the Materials section, and save implications for the Discussion section.

As suggested by the reviewer, we have substantially revised the Introduction to remove any references to the STEPP screening tool. Instead, we discuss risk factors for PTSD more conceptually in this section. We also moved any implications to the Discussion section.

  1. Methods: Provide more information on the STEPP items and how they are endorsed and scored.

As suggested by the reviewer, we included the items from the STEPP screening tool as well as additional information about coding within the section Assessment of Risk Factors for PTSD.

  1. Methods: Add information about Injury Severity Score, including how variable is measured/rated, who reports this information, and whether it is a continuous variable.

We have added a section describing the Injury Severity Score to the Materials and Methods section after the description of Length of Stay.

  1. Results: Replace boy/girl with male/female.

This change has been made throughout the results as well as tables except where “Girl” is referring to the items within the STEPP (since this is the STEPP’s original language).

  1. Results: Reword child/parent total to child/parent STEPP total to increase clarity.

We have made this revision throughout the manuscript.

  1. Results: Clarify what is meant by peds vs. auto.

We have clarified this in the manuscript.

  1. Results: Page 4, line 173, please describe the rationale for using a t-test to assess relationships with age, as is a continuous variable and you lose variability this way.

We wanted to examine age as a categorical variable for older and younger children in light of evidence for age differences in pain experiences (e.g., older children experience more chronic pain and pain persistence; King et al., 2011; Martin et al., 2007). We are open to modifying these analyses if the reviewer has a suggestion for doing so.

  1. Results: Categorize marginal finding for race as non-significant (p>.05) and exclude t values for non-significant results within the sex, age, and race differences section.

These changes have been made within the manuscript.

  1. Results: Confirm that point-biserial correlations were used in correlation table (Table 2) and give rationale for not using chi-squared analyses for STEPP items (dichotomous variables).

We confirm that point-biserial correlations were used in correlation table for the continuous/dichotomous variables. Chi-square analyses were not used to examine the relationship between STEPP items because it was already assumed that these items are related and not independent (they are items from the same measure). Moreover, computing the Pearson correlation for two dichotomous binary variables (which is what was done with the STEPP items in the table) is the same as the Phi, which is the correlation between two dichotomous variables. We are open to modifying or cutting parts of this table if the reviewer feels this is necessary.

  1. Results: Reword sentence after “Hierarchical Regression Analyses” to improve clarity.

Thank you for the suggestion. We have made this change.

  1. Results: Explain rationale for removing pain score and threat to life appraisal from the analyses.

We wanted to explore whether any parental items had any contribution to the analyses when parental items were investigated separately from child items or acute pain intensity as predictors. This was only done for one set of analyses to examine parental items.

  1. Discussion: Temper discussion of implications to the scope of study by distinguishing between direct implications and future directions.

We have made substantial edits throughout the discussion to clarify and limit the implications of our findings and distinguish them from future directions (see Point #1 for additional discussion of this point).

  1. Discussion: Acknowledge there are likely many other variables contributing to LOS that are not examined here. 

We added a point in the limitations addressing this (second-to-last paragraph).

Overall, we appreciate the thoroughness of the reviewer’s suggestions and we hope that, with these changes, this reviewer will find our manuscript suitable for publication within Children’s.

Round 2

Reviewer 1 Report

As my goal here was not to insult the authors nor stop the publication, after substantial clarification and reparation of the manuscript, I see no reason not to grade it as passable and good to publish.

My congratulations to authors.

Best regards